# Clinical Phenotype and Management of Severe Neurotoxicity Observed in Patients with Neuroblastoma Treated with Dinutuximab Beta in Clinical Trials

**DOI:** 10.3390/cancers14081919

**Published:** 2022-04-10

**Authors:** Aleksandra Wieczorek, Carla Manzitti, Alberto Garaventa, Juliet Gray, Vassilios Papadakis, Dominique Valteau-Couanet, Katarzyna Zachwieja, Ulrike Poetschger, Ingrid Pribill, Stefan Fiedler, Ruth Ladenstein, Holger N. Lode

**Affiliations:** 1Pediatric Hematology Oncology, Jagiellonian University Medical College, 30-663 Krakow, Poland; a.wieczorek@uj.edu.pl; 2Oncology Unit, IRCCS Istituto Giannina Gaslini, 16147 Genova, Italy; carlamanzitti@gaslini.org (C.M.); albertogaraventa@gaslini.org (A.G.); 3Centre for Cancer Immunology, University of Southampton, Southampton SO16 6YD, UK; jcgray@soton.ac.uk; 4Department of Pediatric Hematology-Oncology, Agia Sofia Children’s Hospital, 11527 Athens, Greece; vpapadak@otenet.gr; 5Children and Adolescent Oncology Department, Gustave Roussy, 94805 Villejuif, France; dominique.valteau@gustaveroussy.fr; 6Department of Pediatric Nephrology and Hypertension, Jagiellonian University Medical College, 30-663 Krakow, Poland; katarzyna.zachwieja@gmail.com; 7Department for Studies and Statistics and Integrated Research, Children’s Cancer Research Institute, 1090 Vienna, Austria; ulrike.poetschger@ccri.at (U.P.); ingridpr@gmx.at (I.P.); 8Department for Studies and Statistics and Integrated Research at the Children’s Cancer Research Institute of the St. Anna Children’s Hospital and Department of Paediatrics, Medical University, 1090 Vienna, Austria; stefan.fiedler@stanna.at (S.F.); ruth.ladenstein@ccri.at (R.L.); 9Pediatric Hematology and Oncology, University Medicine Greifswald, 17475 Greifswald, Germany

**Keywords:** neuroblastoma, anti-GD2 antibody, dinutuximab beta, neurotoxicity

## Abstract

**Simple Summary:**

Neurotoxicity is an on-target side effect of GD2-directed immunotherapy due to the expression of GD2 on healthy cells. Patients with high-risk neuroblastoma who receive treatment with anti-GD2 immunotherapy, therefore, require close observation and supportive management to improve treatment tolerance and avoid the persistence of neurological symptoms. This study reports on the incidence, clinical course and management of patients who experienced neurotoxicity due to treatment with the anti-GD2 antibody dinutuximab beta, given with or without interleukin-2, in two clinical trials. The majority of severe neurotoxic events were observed in patients treated with dinutuximab beta combined with interleukin-2, with most patients recovering following supportive management. Given the increased risk of neurotoxic events and the lack of clinical benefit reported for the combination treatment in clinical trials, adding interleukin-2 to dinutuximab beta therapy is not recommended. The clinical experiences described here may aid clinicians in managing neurotoxicity associated with dinutuximab beta more effectively.

**Abstract:**

Neurotoxicity is an off-tumour, on-target side effect of GD2-directed immunotherapy with monoclonal antibodies. Here, we report the frequency, management and outcome of patients enrolled in two prospective clinical trials who experienced severe neurotoxicity during immunotherapy with the anti-GD2 antibody dinutuximab beta (DB) administered as short-term infusion (HR-NBL1/SIOPEN study, randomisation R2, EudraCT 2006-001489-17) or as long-term infusion (HR-NBL1/SIOPEN study, randomisation R4, EudraCT 2006-001489-17 and LTI/SIOPEN study, EudraCT 2009-018077-31), either alone or with subcutaneous interleukin-2 (scIL-2). The total number of patients included in this analysis was 1102. Overall, 44/1102 patients (4.0%) experienced Grade 3/4 neurotoxicities (HR-NBL1 R2, 21/406; HR-NBL1 R4, 8/408; LTI study, 15/288), including 27 patients with severe neurotoxicities (2.5%). Events occurred predominantly in patients receiving combined treatment with DB and scIL-2. Neurotoxicity was treated using dexamethasone, prednisolone, intravenous immunoglobulins and, in two patients, plasmapheresis, which was highly effective. While neurological recovery was observed in 16 of 21 patients with severe neurotoxicities, 5/1102 (0.45%) patients experienced persistent and severe neurological deficits. In conclusion, severe neurotoxicity is most commonly observed in patients receiving DB with scIL-2. Considering the lack of clinical benefit for IL-2 in clinical trials so far, the administration of IL-2 alongside DB is not recommended.

## 1. Introduction

The promising results of clinical trials with anti-GD2 monoclonal antibodies in patients with high-risk neuroblastoma led to the introduction of immunotherapy as the standard of care in the maintenance phase of first-line treatment settings [1,2,3,4,5,6]. Immunotherapy with anti-GD2 antibodies is administered either as monotherapy or in combination with cytokines, following induction chemotherapy and consolidation therapy with high-dose chemotherapy and autologous stem cell rescue [1,3]. In the ANBL0032 study of the Children’s Oncology Group (COG), treatment with dinutuximab, a human/mouse chimeric anti-GD2 antibody produced in SP2/0 cells (ch14.18) [7], resulted in improved survival in patients with high-risk neuroblastoma when given in combination with granulocyte-macrophage colony-stimulating factor (GM-CSF) and intravenous (i.v.) interleukin-2 (IL-2) in alternating cycles [3]. In addition, two clinical trials conducted by the International Society of Paediatric Oncology European Neuroblastoma Group (SIOPEN) also demonstrated a benefit for patients with high-risk neuroblastoma treated with dinutuximab beta (DB), a similar anti-GD2 antibody produced in Chinese hamster ovary (CHO) cells (ch14.18/CHO) [1,2,8,9,10]. Short-term (8 h for 5 days) or long-term infusion (LTI; continuous infusion over 10 days) of DB with or without subcutaneous IL-2 (scIL-2) improved survival in the first-line maintenance setting (HR-NBL1/SIOPEN study) [1,2,8] as well as in patients with relapsed and refractory neuroblastoma (LTI/SIOPEN study) [9,10]. Although effective, the therapy is commonly associated with significant side effects, such as neuropathic pain, capillary leak syndrome and allergic reactions. Most of the side effects can be managed using appropriate prophylaxis and adequate supportive treatment during immunotherapy [11].

Neurotoxicity has also been reported in association with anti-GD2 immunotherapy [1,8,11]. While it is a relatively uncommon side effect, it specifically relates to the target antigen GD2 [1,6,8,11]. Although GD2 is a well-recognised tumour-associated antigen, its expression is not restricted to neuroectodermal tumours such as neuroblastoma, but can also be found on healthy tissues, in particular those of neuronal origin [7,12]. The expression of GD2 in the central nervous system (CNS) and the peripheral nervous system (PNS) results in on-target/off-tumour side effects when GD2-directed immunotherapies are administered [6,13,14]. One well known off-tumour side effect of anti-GD2 antibody therapy is the induction of neuropathic pain [11]. In animal models, which approximate the pain associated with anti-GD2 antibodies in humans in terms of timing and quality, anti-GD2-specific binding to Aδ and C pain fibers results in decreased mechanical stimulus thresholds [15]. Therefore, clinical use of anti-GD2 antibody therapy requires intensive co-administration of analgesic drugs, including i.v. morphine, to make this treatment tolerable [11]. 

On-target/off-tumour side effects associated with the expression of GD2 in the CNS may be more severe than in the PNS. In particular, the occurrence of transverse myelitis associated with paraplegia has been described in patients receiving anti-GD2 antibody therapy [11,16]. Given that anti-GD2 antibody therapy for high-risk neuroblastoma has been developed in combination with cytokines, in particular IL-2 [1,3], it is important to note that uncommon, but severe, neurotoxicity has also been reported for IL-2 therapy alone, including coma, convulsions and paralysis (≥1/1000 to <1/100 patients) [17]. Other side effects include changes in mental status, ataxia, and blindness, with evidence of demyelination as a radiological finding [17,18,19]. There are also reports of permanent neurological defects. More common side effects associated with IL-2 therapy are dizziness, paresthesia and somnolence, followed by neuropathy and lethargy [17]. 

Here, we report the incidence, clinical course and management of severe neurotoxicity observed in patients with high-risk neuroblastoma treated with DB with and without scIL-2 in two prospective SIOPEN trials (LTI study and HR-NBL1 study Randomisation 2 and 4 (R2 and R4)).

## 2. Materials and Methods

### 2.1. Patients

The occurrence and management of neurotoxicity was assessed in patients with high-risk neuroblastoma enrolled in two prospective SIOPEN studies: HR-NBL1/SIOPEN [1,8] and LTI/SIOPEN [10]. This analysis includes all 814 patients of the R2 and R4 phases of the HR-NBL1/SIOPEN study and all 288 patients enrolled in the LTI/SIOPEN study.

### 2.2. Study Designs

The LTI/SIOPEN study (EudraCT 2009-018077-31) is a prospective Phase II trial in patients with relapsed and refractory high-risk neuroblastoma (Figure 1A,B) [10]. The trial started as a single-arm study, where patients were treated with DB in combination with scIL-2, and was later amended to a randomised study, where patients received DB with or without scIL-2 [10]. The treatment regimen in the LTI/SIOPEN study included scIL-2 given once a day for five days (Days 1–5, 6 × 106 IU/m^2^/day), followed by combined administration of scIL-2 once a day (Days 8–12, 6 × 106 IU/m^2^/day) and a 10-day continuous i.v. infusion of DB (Days 8–17 10 mg/m^2^/day) [10]. Oral isotretinoin was given for 2 weeks following DB infusion (Days 19–32, 160 mg/m^2^/day) [10]. Patients received up to five 35-day treatment cycles [10]. In the randomised phase of the trial, the treatment schedule for DB, scIL-2 and isotretinoin remained unchanged but was tested against a treatment arm without scIL-2 [10]. 

The HR-NBL-1/SIOPEN study (EudraCT 2006-001489-17) is a prospective Phase III trial in patients with newly diagnosed high-risk neuroblastoma and includes four completed randomisations (R1–4), which are described elsewhere [1,8,20,21,22]. In the randomisation phase 2 (R2, Figure 1C,D), patients received DB as short-term infusion (STI) consisting of 20 mg/m^2^/day given as 8 h infusion on 5 consecutive days (Days 8–12) either with or without scIL-2 (Days 1–5 and Days 8–12; 6 × 10^6^ IU/m^2^/day) [1]. Oral isotretinoin was given for 2 weeks (Days 15–28; 160 mg/m^2^/day). All patients received up to five 35-day cycles [1]. In the randomisation phase 4 (R4, Figure 1E,F), the infusion schedule of DB was changed to 10 days continuous infusion (Days 8–17; 10 mg/m^2^/day), with the same cumulative dosage of 100 mg/m^2^ per cycle as the STI [8]. The total dose of scIL-2 was reduced to 30 × 10^6^ IU/m^2^/cycle, which was administered in a relaxed schedule during the combination treatment with DB (Days 1−5, 8, 10, 12, 14; 16; 3 × 10^6^ IU/m^2^/day) [8]. The cumulative dosage of isotretinoin remained the same, but it was given on Days 19–32 of the treatment cycle [8].

Both trials stopped recruitment, and results of HR-NBL1-R2 were reported by Ladenstein et al. [1] The LTI study and the R4 phase of HR-NBL1 are currently in the follow-up phase awaiting data maturity to report activity and efficacy endpoints, with early results communicated in 2019 [8,10].

### 2.3. Assessments

All data on Grade 3/4 adverse events classified as neurotoxicity according to the Common Terminology Criteria for Adverse Events (CTCAE) were collected from both trials and analysed. A subgroup of patients who experienced severe CNS neurotoxicity underwent more detailed analysis. Criteria for severe CNS neurotoxicity are shown in Table 1. For the purpose of this report, we focused on severe CNS neurotoxicity with substantial and prolonged neurological deficits observed among patients with Grade 3/4 toxicity and causal relationship to DB. Severe DB-related CNS neurotoxicity was defined by the occurrence of clinical symptoms of the CNS without any other detectable reason. If radiological imaging was performed, results had to be consistent with the clinical findings, including the presence of CNS inflammation and/or demyelination (Table 1). Patients with severe pain and ophthalmoplegia occurring without any additional neurological sensory or motor dysfunction were not included as severe CNS neurotoxicity in this report.

## 3. Results

Overall, 44 of the 1102 patients (4.0%) included in the two SIOPEN trials experienced Grade 3/4 neurotoxicities, with 27/1102 (2.45%) patients fulfilling the criteria of severe CNS neurotoxicity (Table 1).

The majority of patients recovered, namely, 33 of 38 (86.8%), including 16 patients presenting with severe CNS neurotoxicity; data were not available for six patients. Only 5/1102 (0.45%) patients presented with persistent and severe neurological deficits. The distribution and outcomes of all patients with Grade 3/4 neurotoxicity are shown in Table 2 and the clinical characteristics and management of all patients with severe CNS neurotoxicity are shown in Table 3. Case descriptions and therapeutic interventions are provided in the Appendix A.

The most common severe neurological side effects observed were paresis or hypotonia (*n* = 10), neurogenic bladder (*n* = 7), seizures (*n* = 6), ataxia and/or gait disturbances (*n* = 4) and cranial nerve palsy (*n* = 3). Patients also reported pain, sensory loss, somnolence, mood and behavioural changes, and visual disturbances, including one patient with blindness. In 12 patients, neurotoxicity presented with more than one symptom. The most common single symptoms were seizures and cranial nerve disturbances, all of which resolved without sequelae. No patient with severe neurotoxicity had meningeal or CNS involvement from their neuroblastoma.

### 3.1. LTI/SIOPEN Study

In the LTI/SIOPEN study, 15 out of 288 patients (5.2%) reported Grade 3/4 neurotoxicity (Table 2), including 10 patients with severe CNS neurotoxicity (3.5% of all LTI patients), 3 of whom experienced persistent neurological deficits (1.0% of all LTI patients). All events in this study occurred in patients treated with DB plus scIL-2. No neurotoxic events were observed in the first week of scIL-2 (Days 1–5 of the cycle) or during therapy with DB alone. There was no correlation between the disease status at study entry and the incidence of severe neurotoxicity (patients with neurotoxiciy: 60% not in complete response (non-CR) vs. 40% in complete response (CR), patients without neurotoxicity: 60% non-CR vs. 40% CR; *p* = 10,000; Fishers exact test). The serum levels of DB determined at the occurrence of symptoms did not exceed expected values of 12.56 ± 0.68 µg/mL [23]: mean concentration of 7.0 µg/mL ± 1.9 µg/mL (range 2.0–14.9 µg/mL) (Table 3). In two cases where samples were available, proinflammatory cytokines interleukin-6 (IL-6) and interferon-ɣ (IFN-ɣ) in peripheral blood were increased. Therapy with DB was stopped in all patients with severe neurotoxicity and six patients received immunosuppressive treatment.

Cerebrospinal fluid (CSF) was analysed in 5/10 patients with severe neurotoxicity. There were no relevant findings in routine parameters of CSF except for a distinct increase in the protein level (albumin) in four patients (0.55–1.8 g/L), which might be explained by a treatment-associated disturbance of the blood–brain barrier. The leukocyte number was only slightly increased (from 2 to 26 cells/µL). The concentration of DB in the CSF was determined in three patients, with levels of ≤0.02 µg/mL detected. Viral examinations, including polymerase chain reaction (PCR) for varicella zoster virus (VZV), were negative in all evaluated cases. Interestingly, 3/10 patients had VZV or herpes simplex virus (HSV) infection during or shortly before the DB therapy. 

In 10 patients with available magnetic resonance imaging (MRI) scans, radiological review revealed inflammatory CNS lesions (Figure 2). One of these patients was diagnosed with cytotoxic brain stem oedema (Figure 2A) and five patients demonstrated demyelination, including four with transverse myelitis (Figure 2B–D) and 1 with demyelinating neuropathy involving the dorsal roots of the cauda equina. MRI scans were performed in a timely manner (within days) after the onset of clinical symptoms. More details on these cases are provided in the Appendix A.

### 3.2. HR-NBL1/SIOPEN Study (R2 and R4)

In the R2 phase of the HR-NBL1/SIOPEN study (*n* = 406), 21 patients (5.2%) experienced Grade 3/4 neurotoxicity (14 central and 7 peripheral events), including 9 patients with severe CNS neurotoxicity (Table 2 and Table 3).

Of the 14 patients with central neurotoxicities, 3 patients were treated with DB alone and 11 were treated with DB in combination with scIL-2. Initially, only eight patients were classified as severe CNS neurotoxicity; two presented with posterior reversible encephalopathy (PRES), four with seizures, one with mood disturbances followed by motor weakness, photophobia and taste impairment; and one with toxic demyelinating encephalopathy (paresis and coma) who did not fully recover. All severe neurotoxicities occurred in the group of patients treated with DB and scIL-2, except for one patient who experienced PRES while treated with DB alone. 

Of the seven patients with Grade 3/4 peripheral neurotoxicity (paraesthesia and deficits in motor function), six were in the group of patients treated with DB plus scIL-2 and one in the group treated with DB alone. One patient treated with the combination therapy was classified as peripheral neuropathy, despite the occurrence of tetraparesis. The patient also experienced urinary retention and mydriasis. Based on the clinical symptoms, it is likely that this patient had transverse myelitis, and was not assigned to the correct neurotoxicity category. The patient did not recover, and was finally classified as having severe CNS neurotoxicity. Overall, 9 of 406 (2.2%) patients in the R2 phase of the HR-NBL1 study presented with severe CNS neurotoxicity, with 2 patients experiencing persistent neurological deficits (0.5% of all R2-HR-NBL1 patients).

In the R4 phase of the HR-NBL1 study (*n* = 408), eight (2%) patients presented with Grade 3/4 neurotoxicities (four central and four peripheral events), with five reported in the DB only group and three in the DB plus scIL-2 group. In two patients, CNS inflammatory lesions were identified on radiological imaging, with one patient demonstrating normal CSF results. All eight patients recovered without sequelae. HR-NBL1 study patients (R2 and R4 combined) who were not in complete remission (non-CR) before immunotherapy showed a trend towards a higher incidence of severe neurotoxicity (patients with severe neurotoxicity: 86% non-CR vs. 15% CR, patients without neurotoxicity: 55% non-CR vs. 45% CR; *p* = 0.0628; Fishers exact test).

In summary, we observed 44 out of 1102 patients with Grade 3/4 neurotoxicities: 17/814 patients (2%) in the front-line setting and 27 of 288 patients in the relapse/refractory setting (9.4%). The majority of Grade 3/4 neurotoxicities (35/44; 79.5%) occurred in patients treated with DB plus scIL-2. Recovery was observed in 33/38 patients (86.8%), which was achieved with different, non-standardised interventions, including the use of dexamethasone, prednisolone, i.v. immunoglobulins (IVIGs) and plasmapheresis. A total of 12 patients received immunomodulatory treatment combinations of IVIGs and/or steroids, and in 2 patients, who did not respond to this treatment, plasmapheresis resulted in the complete resolution of symptoms (Appendix A). However, 5/44 patients (11.4%) demonstrated persisting severe neurological deficits.

## 4. Discussion

Here, we report the occurrence and management of severe neurotoxicity experienced by patients with high-risk neuroblastoma who received treatment with DB with or without scIL-2 in the front-line or relapse/refractory setting within two prospective clinical trials.

Neurotoxicity is a side effect of anti-GD2 antibodies occurring irrespective of their origin (mouse, chimeric, humanised). Neuropathic pain, confusion and sensorimotor neuropathy caused by demyelination have been described in a Phase I study investigating the murine monoclonal anti-GD2 antibody 14G2a in patients with metastatic melanoma [24]. Similarly, a Phase I study with the human/mouse chimeric anti-GD2 antibody ch14.18 reported optic nerve atrophy in two patients with neuroblastoma who also received radiotherapy [25]. In the prospective COG trial in children with high-risk neuroblastoma, patients randomised to dinutuximab (ch14.18 produced in SP2/0 cells [7]) in combination with i.v. IL-2 and GM-CSF, CNS toxicity was reported in 6 of 137 patients (4.4%) and manifested as encephalopathy, confusion, psychosis and seizure [3]. Neurotoxicity was also reported in the preceding Phase I study of dinutuximab in combination with GM-CSF in children with neuroblastoma [26]. In addition, treatment with the murine anti-GD2 antibody 3F8 was reported to be associated with neurotoxicity, with 5 of 215 patients (2.3%) developing PRES [27]. There was no case of transverse myelitis in this study, which might be explained by the absence of concomitant IL-2 therapy.

Transverse myelitis has, however, been reported in three patients treated with dinutuximab therapy [16]. Clinical symptoms were similar to those observed in our analysis and included weakness of bilateral lower extremities, urinary retention and progression to paraplegia [16]. Two of these patients were treated with concomitant IL-2 and GM-CSF, while one patient received dinutuximab combined with chemotherapy [16]. All symptoms resolved after immunomodulatory treatment with steroids and/or IVIG and/or plasmapheresis [16]. As all patients experienced neurotoxicity during consecutive courses of dinutuximab, the authors suggested that the previous exposure to the drug induced an anti-idiotypic immune response with crossreactivity against neuronal tissue [16]. However, in our analysis, neurotoxicity occurred in the first treatment cycle and most of these patients did not receive previous anti-GD2 immunotherapy, suggesting that other mechanisms may be involved.

The pathogenesis of the observed neurotoxicity is not entirely clear. This “on-target, off-tumour” side effect is not observed with human mouse chimeric monoclonal antibodies that target other antigens, such as CD20 (e.g., rituximab) or tumour necrosis factor-alpha (e.g., infliximab). Thus, neurotoxicity is associated with antibodies binding to GD2 followed by the induction of an inflammatory response against GD2-expressing neuronal tissue [28]. Since anti-GD2 antibodies are often used in combination with cytokines, the neurotoxic inflammation may be further enhanced, in particular in the presence of IL-2 [1,3,11]. As mentioned earlier, severe neurotoxicity is a reported side effect of IL-2 monotherapy including neuropathy, coma, convulsions, paralysis and leukoencephalopathy [17]. However, in both trials analysed in this report, neurotoxicity was not observed during the first five days of the monotherapy period with scIL-2, except for one patient who had an event during the first five days of Cycle 2, which might be due to carry-over exposure to DB from the previous cycle. Therefore, the anti-GD2 antibody appears to be the major factor for this neurotoxicity to occur. The frequency of severe neurotoxicity is very low in patients treated with DB alone, which indicates that IL-2 may be largely responsible for increasing the frequency and amplitude of this side effect, potentially by enhancing the inflammation triggered by the antibody effector function. The observation of neurotoxicity as a side effect of treatment with anti-GD2 antibodies is not new and appears to be a specific effect of anti-glycolipid antibodies [29,30]. Various antibodies directed against more than 20 different glycolipids have been associated with acute and chronic neuropathy syndromes [30]. 

Our analysis revealed a substantial number of patients with severe CNS neurotoxicities who had a reactivation/infection with the herpes viruses VZV and HSV (3/10 patients in the LTI study). However, it is not clear if there is a causal relationship. The herpes viruses family is known to exhibit neurotropism; VZV in particular is known to cause myelitis, even without involvement of the skin (zoster sine herpete) [31,32]. Myelitis might also be the only presentation of VZV infection, and examination of CSF in such cases may show normal results [32]. However, not all patients with HSV or VZV infection may present with neurotoxicities. In our analysis, there were 2 reported patients with active infections who received DB combined with scIL-2 and did not exhibit any neurotoxic side effects. Considering the poor prognosis for patients with high-risk neuroblastoma, the diagnosis of HSV or VZV infection should not exclude patients from receiving DB. The infection should be treated first until complete resolution, and acyclovir prophylaxis should be given during treatment with DB.

Since the neurotoxic effect of DB treatment is not observed with other chimeric human mouse monoclonal antibodies, it is likely to be associated with the GD2 specificity of the antibody. Moreover, the observation that removing DB by plasmapheresis led to a reduction in symptoms in two patients, and that the attempt of therapy reintroduction caused recurrence of symptoms in two patients, suggests that there is a causal relationship between DB and neurotoxicity. However, it is surprising that we did not find DB in the CSF of these three patients with neurotoxicity, suggesting that these events might also be due to indirect bystander effects of the immunological activation. In addition, the serum concentrations of DB were also not unusually high compared to the expected maximum concentration of 12.56 ± 0.68 µg/mL [23], indicating that high DB levels are unlikely to be responsible for the severe events observed. As mentioned earlier, IL-2 may be the primary driver of severe neurotoxicity in these patients. 

In patients with neurotoxicity reported in our analysis, temporary immunosuppression with steroids (prednisolone and dexamethasone) or IVIG in standard doses was initiated immediately after the occurrence of symptoms, resulting in recovery in the majority of patients. In two patients who did not improve after immunosuppression, plasmapheresis was effective. One of these cases has recently been reported [33]. Although increased proinflammatory cytokine levels were found in some cases, other immunosuppressive drugs such as anti-IL-6 monoclonal antibodies were not administered due to lack of experience with these. As the recurrence of neurological symptoms has been described in three out of seven patients after the attempt of drug rechallenge, DB therapy should be discontinued permanently after the occurrence of severe neurotoxicity.

Based on our experience, we suggest the following management in case of neurological symptoms during immunotherapy (Table 4): Once symptoms occur, it is crucial to start immunosuppression with dexamethasone or prednisolone and intravenous immunoglobulin immediately. In parallel, other causes for neurological symptoms need to be ruled out, including infection and disease progression. Therefore, immediate imaging studies and CSF analysis are important steps. Until an infectious cause is excluded, antimicrobial and antiviral treatment is advisable. Taking into consideration a possible influence of HSV/VZV infection/reactivation in the pathogenesis of myelitis, even in the absence of evident skin lesions, viral diagnostics by PCR for VZV and HSV should be performed using blood and CSF. Acyclovir treatment should be considered at least until PCR results come back negative, even if no evidence of VZV/HSV is found in laboratory examinations. Furthermore, special care should be taken when DB therapy is initiated in patients who had active VZV or HSV infections shortly before planned treatment. Based on the cases reported here, it is not possible to recommend a clear time frame between infection and the start of therapy, but a complete resolution of active HSV/VZV infection should be achieved before starting DB. In addition, early intervention is recommended to reduce the severity and duration of neurological symptoms and prevent persistent disability. Follow-up examinations are recommended in patients with persistent symptoms and/or initial MRI findings upon the onset of neurotoxicity every 4–6 weeks to determine the course and make a decision about the escalation or de-escalation of therapy. 

## 5. Conclusions

Severe neurotoxic events observed in the two SIOPEN trials occurred more frequently in patients receiving DB combined with scIL-2 than in those receiving DB monotherapy. Although supportive management resulted in recovery in the majority of patients, a very small number of patients with severe CNS neurotoxicities demonstrated persistent neurological deficits, which are likely due to the co-administration of scIL-2. Since scIL-2 has not shown any clinical benefit when added to DB therapy [2,3,4], the findings reported here provide further evidence that scIL-2 should be omitted from the treatment regimen. When using DB therapy, it is crucial to strictly follow the patients to diagnose symptoms of neurotoxicity early, stop DB treatment and start immunosuppression. In patients who experience severe neurotoxicity, DB treatment should be discontinued permanently.

## Figures and Tables

**Figure 1 cancers-14-01919-f001:**
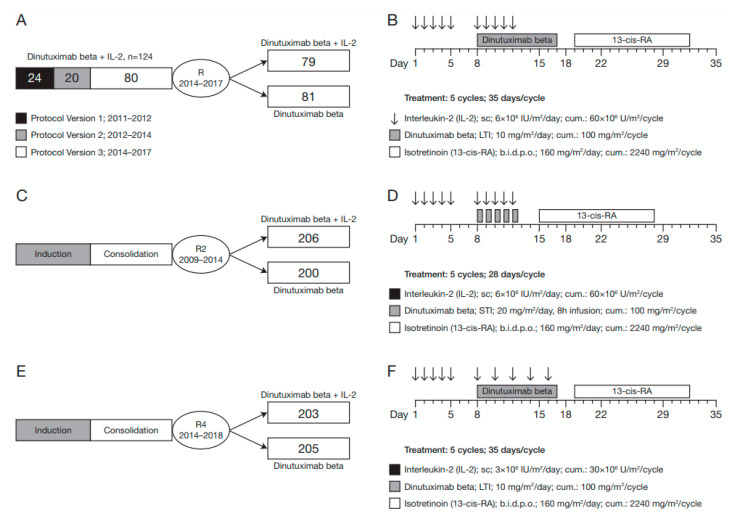
Schematic overview of study designs and treatment schedules in the LTI/SIOPEN study and the HR-NBL1/SIOPEN study. (**A**,**B**) Study design and treatment schedule of the LTI study. The study was initiated as a single-arm study of DB combined with scIL-2 in patients with relapsed/refractory high-risk neuroblastoma and amended in 2014 to include a randomizat design. Patients received either DB alone or DB combined with scIL-2. DB was administered as LTI (10 mg/m^2^ continuous infusion over 10 days; total dose 100 mg/m^2^). (**C**,**D**) Study design and treatment schedule of HR-NBL1-R2. Newly diagnosed patients with high-risk neuroblastoma were randomizat in the maintenance treatment phase to receive either DB alone or DB combined with scIL-2. DB was administered as STI (20 mg/m^2^/day on 5 consecutive days, 8 h infusions; total dose 100 mg/m^2^). (**E**,**F**) Study design and treatment schedule of HR-NBL1-R4. The study was amended to evaluate LTI of DB and a dose-reduced relaxed schedule of scIL-2. B.i.d.p.o., twice-daily oral administration; DB, dinutuximab beta; LTI, long-term infusion; R, randomization; RA, retinoic acid; scIL-2, subcutaneous interleukin-2; STI, short-term infusion.

**Figure 2 cancers-14-01919-f002:**
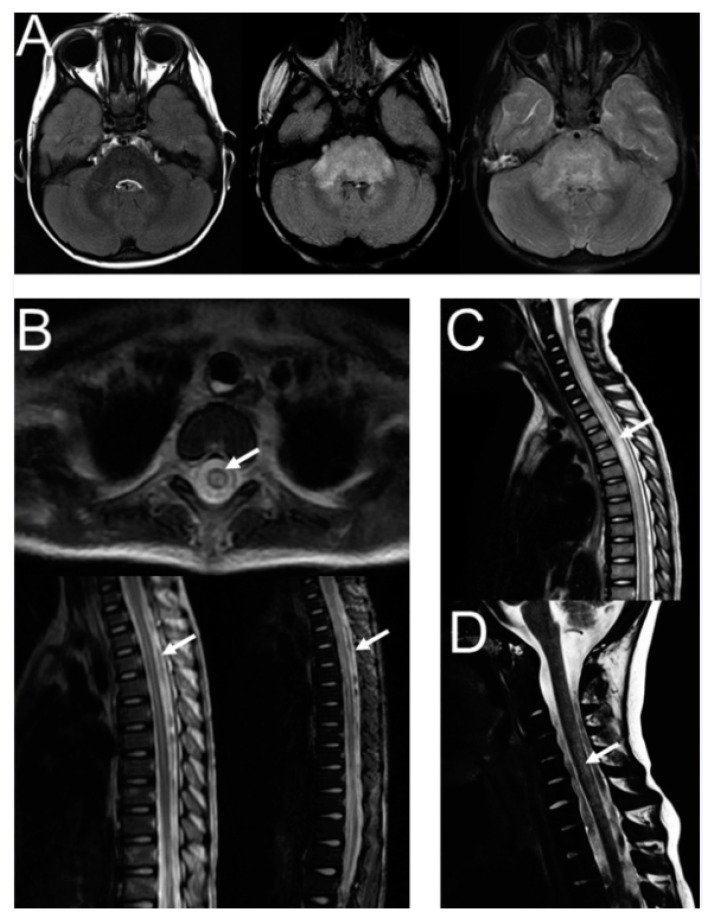
Imaging results of patients with severe CNS neurotoxicities during immunotherapy with DB and scIL-2. (**A**) Patient 1. MRI of brain showing inflammation and oedema of mesencephalon, pons and medulla oblongata. (**B**) Patient 3. Frontal and sagittal MRI of spinal cord, with hyperintensive zone in STIR- and T2-weighted images. (**C**,**D**) Patient 2 and 4, respectively. Sagittal MRI of spinal cord, with hyperintensive zone in T2-weighted images. Arrows indicate areas of inflammatory response. DB, dinutuximab beta; CNS, central nervous system; MRI, magnetic resonance imaging; scIL-2, subcutaneous interleukin-2; STIR, short-TI inversion recovery.

**Table 1 cancers-14-01919-t001:** Criteria of severe CNS neurotoxicity.

Clinical Symptoms of CNS Dysfunction	Radiological Signs
Muscle paresis and/or paraplegia	CNS inflammation and/or demyelination
Urinary retention/neurogenic bladder not associated with morphine use	
Coma	
Seizures	
Cranial nerve palsy	
Ataxia	

CNS, central nervous system.

**Table 2 cancers-14-01919-t002:** Distribution and outcome of patients with severe neurotoxicity in the LTI and HR-NBL1 (R2 and R4) study.

Patients, *n*	LTI Study	HR-NBL1 (R2)	HR-NBL1 (R4)	Total
**Enrolled**	288	406	408	1102
**Gr 3/4 neurotoxicity ***	15	21	8	44
**with DB**	0	4	5	9
**with DB + scIL-2**	15	17	3	35
**Severe CNS neurotoxicity ^#^**	10	9	8	27
**with DB**	0	1	5	6
**with DB + scIL-2**	10	8	3	19
**Recovery**	10	17	6	33 ^†^
**Persistent severe neurotoxicity**	3	2	0	5

* All neurotoxicity events reported according to CTCAE. ^#^ Severe neurotoxicity according to the definition outlined in Table 1; ^†^ Data were missing for 6 patients with severe CNS neurotoxicity. DB, dinutuximab beta; CNS, central nervous system; CTCAE, Common Terminology Criteria for Adverse Events; Gr, grade; R, randomisation; scIL-2, subcutaneous interleukin-2.

**Table 3 cancers-14-01919-t003:** Clinical characteristics and management of severe CNS neurotoxicity in the LTI study and the HR-NBL1 study.

Pt	Study	Schedule	Time of Onset	Symptoms	MRI Findings	CSF	HSV/VZV	DB Level (Serum) *	DB Level (CSF)	Treatment	Rechallenge with DB	Symptom Resolution	MRI Follow-Up
1	LTI	DB LTI + scIL-2	C1/D13	Dysphagia Apnoea Paraparesis Fixed pupils	Cytotoxic oedema in brain stem	ND	No	2.5 µg/mL	ND	IVIG Steroids Ventilation	No	No (minimal improvement)	NA
2	LTI	DB LTI + scIL-2	C1/D14	Back pain Neurogenic bladder Paraplegia Sensory loss	Myelitis (thoracic to lumbar)	Protein 1.8 g/L Glc 3.7 mmol/L Leucocytes 26/µL Oligoclonal band	Yes (thoracic shingles 6 wks before therapy)	4.3 µg/mL	ND	IVIG Steroids	No	No	Partial resolution after 8 months
3	LTI	DB LTI + scIL-2	C1/D15	Paraplegia Neurogenic bladder Pain Hyposthenia of upper extremities Diaphragm paresis	Myelitis (thoracic)	Protein 0.96 g/L Glc 3 mmol/L Leukocytes 4/µL VZV PCR neg	Yes (skull skin shingles after the first round of scIL-2 during C1/D1–D5)	5.8 µg/mL	ND	IVIG Steroids Plasmapheresis	No	Yes	Complete resolution
4	LTI	DB LTI + IL-2	C3/D17	Ataxia Tetraparesis Pain Paresthesia	Myelitis (thoracic)	Protein 0.55 g/L Leukocytes 2/µL	Yes (HSV 3 wks before DB)	14.8 µg/mL	0.02 µg/mL	IVIG Steroids	No	Yes	Complete resolution
5	LTI	DB LTI + scIL-2	C1/D15	Urinary retention Hyposthenia Sensorimotor demyelinating polineuropathy	Demyelinating neuropathy of dorsal roots	Albumino-cytologic dissociation	No	4.6 µg/mL	<0.01 µg/mL	IVIG	Yes (symptoms reoccurred after 4 h at 50% dose)	Yes	NA
6	LTI	DB LTI + scIL-2	C1/D9	Blurred vision, Neurogenic bladder Paraplegia	Myelitis	ND	No	ND	<0.01 µg/mL	Steroids	No	No (persistent neurogenic bladder)	NA
7	LTI	DB LTI + scIL-2	C1/D9	Somnolence Encephalopathy on EEG	ND	ND	ND	ND	ND	No treatment	No	Yes	NA
8	LTI	DB LIT + scIL-2	C2/D11	Urinary retention	ND	ND	ND	2.0 µg/mL	ND	No treatment	No	Yes	NA
9	LTI	DB LTI + scIL-2	ND	Encephalopathy	ND	ND	ND	14.9 µg/mL	ND	ND	ND	ND	ND
10	LTI	DB LTI + scIL-2	ND	Seizures	ND	ND	ND	ND	ND	ND	ND	ND	ND
11	HR-NBL1R2	DB STI + scIL-2	C4/D10	Coma Paresis (no DB given in the cycle)	Encephalomyelitis	ND	ND	ND	ND	Steroids	No	No	NA
12	HR-NBL1R2	DB STI	C2/D7	6th cranial nerve palsy	PRES	ND	ND	ND	ND	Steroids	Yes (no recurrence)	Yes	Complete resolution
13	HR-NBL1R2	DB STI + scIL-2	C2/D3	Seizures Ondine syndrome	ND	ND	ND	ND	ND	Antibiotics (merpenem, vancomycin)	ND	Yes	NA
14	HR-NBL1R2	DB STI + scIL-2	C1/D5	Seizures	ND	ND	ND	ND	ND	Midazolam	Yes (no recurrence)	Yes	NA
15	HR-NBL1R2	DB STI + scIL-2	C1/ND	Seizures	ND	ND	ND	ND	ND	ND	Yes (only DB, no scIL-2; no recurrence)	Yes	NA
16	HR-NBL1R2	DB STI + scIL-2	C1/D9	Seizures	ND	ND	ND	ND	ND	ND	ND	ND	NA
17	HR-NBL1R2	DB STI + scIL-2	C1/D9	Mood disturbances Motor weakness/hypotonia Mydriasis Photophobia Taste change	ND	ND	ND	ND	ND	No treatment	ND	Yes (child needs glasses)	NA
18	HR-NBL1R2	DB STI + scIL-2	C1/D15	ND	PRES	ND	ND	ND	ND	ND	ND	ND	NA
19	HR-NBL1R2	DB STI + scIL-2	ND	Paresis Urinary retention Mydriasis (patient initially classified as non- severe peripheral neuropathy)	ND	ND	ND	ND	ND	ND	ND	No	ND
20	HR-NBL1R4	DB LTI + scIL-2	C1/D10	Severe somnolence Hypotonia No reaction to pain	Encephalitis	Normal	ND	ND	ND	IVIG, Steroids	No	Yes	Residual focal lesions
21	HR-NBL1R4	DB LTI	C3/D13	Behavioural change Ataxia Hyperkinesis Gait disturbances Torticollis Blindness	Encephalitis	Protein 0.18 g/L Glc 44 mg/dL	No (blood and CSF)	ND	ND	IVIG, Steroids Plasma- pheresis	No	Yes	ND
22	HR-NBL1R4	DB LTI	C3/D17	Facial paralysis	Discreet infiltration of the acoustic- facial package	Normal	ND	ND	ND	Steroids	Yes (no recurrence)	Yes	ND
23	HR-NBL1R4	DB LTI	C4/ND	Left sided facial palsy	Mucosal thickening of left mastoid cell	ND	ND	ND	ND	Meropenem	ND	ND	ND
24	HR-NBL1R4	DB LTI	C1/D22	Agitation with life threatening behaviour (behavioural disturbances since treatment start)	ND	ND	ND	ND	ND	Hydroxyzinum	Yes (symptoms reoccurred after 90 min of infusion)	Yes	NA
25	HR-NBL1R4	DB LTI	C1/D11	Sensory disturbances in all extremities Gait and fine catch disturbances	Sensory axonal neuropathy	High protein	ND	ND	ND	IVIG Steroids	ND	ND	ND
26	HR-NBL1R4	DB LTI + scIL-2	C1/D15	Seizures	Normal	ND	ND	ND	ND	ND	Yes (no recurrence)	Yes	ND
27	HR-NBL1R4	DB LTI + scIL-2	C1/D27	Dragging left foot Gait disturbances	ND	ND	ND	ND	ND	ND	ND	Yes	ND

* The expected maximum serum concentration of DB at the end of infusion was 12.56 ± 0.68 µg/mL, as previously reported for patients in the LTI study [23]. CNS, central nervous system; C, cycle; CSF, cerebrospinal fluid; D, day; DB: dinutuximab beta; EEG, electroencephalogram; glc, glucose; h, hours; HSV, herpes simplex virus; IVIG, intravenous immunoglobulins; LTI, long-term infusion; mins, minutes; MRI, magnetic resonance spectroscopy; NA, not available; ND, not determined; PCR, polymerase chain reaction; PRES, posterior reversible encephalopathy syndrome; pt, patient; scIL-2, subcutaneous IL-2; STI short-term infusion; VZV, varicella zoster virus, wks, weeks.

**Table 4 cancers-14-01919-t004:** Management recommendations for suspected neurotoxicity during immunotherapy with DB.

Stop DB infusion/immunotherapy
Start antibacterial and antiviral therapy for potential infectious cause
Examine for VZV and HSV infection (PCR of blood and CSF) Rule out other cause for neurological symptoms by cMRI, MRI of spinal cord and evaluate metabolic conditions
EEG if clinically indicated
Carry out CSF examination
Urgently start immunosuppression with dexamethasone or prednisolone and IVIG
In case of no improvement of symptoms, consider plasmapheresis

DB, dinutuximab beta; cMRI, cardiac magnetic resonance imaging; CSF, cerebrospinal fluid; EEG, electroencephalogram; HSV, herpes simplex virus; IVIG, intravenous immunoglobulins; MRI, magnetic resonance imaging; PCR, polymerase chain reaction; VZV, varicella zoster virus.

## Data Availability

The data presented in this study are available on request from the academic sponsor.

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
