# Peer review of "Clinical Phenotype and Management of Severe Neurotoxicity Observed in Patients with Neuroblastoma Treated with Dinutuximab Beta in Clinical Trials"

_cancers, 2022, doi:10.3390/cancers14081919_

Round 1
Reviewer 1 Report
The authors analysed the side effects of the anti GD2-antibody immunotherapy in high-risk neuroblastoma patients. The report is of high interest among paediatric oncologist treating patients with high-risk neuroblastoma. Moreover, the treatment recommendations developed by the authors are of great value for treating physicians. General comment: The paper is well written and easy to read. The conclusions are supported by the data. The only weakness of the analysis it that – although the data were collected within prospective clinical trials - details on presentation, management and outcome of neurotoxic events were not available in all patients.
Minor comments/recommendations:
- Please give the number of patients with meningeal and/or CNS lesions at least among those patients with neurotoxic events.
- Was there any association with the remission status, i.e., are patients in PR more at risk for neurotoxic events?
- According to table 3, 16 patients completely recovered, five patients did not and data were missing in six patients. Therefore, the complete recovery rate should better be given as 16/22 instead of 22/27. The same is true for the patient numbers of the respective trials.
- The layout of table 1 should be revised prior to publication in order to improve readability. The target serum concentration of DB should be given.
- In Table 4, the CSF-examination should be mandatory because CSF is collected for HSV- and HSV-PCR anyway.
Author Response
- Please give the number of patients with meningeal and/or CNS lesions at least among those patients with neurotoxic events.
We have complete information for all 27 patients with severe neurotoxicity regarding their meningeal/CNS involvement and found that none of these patients had CNS disease, and we added a sentence in the results section (line 203). However, we believe that the number of patients with such disease characteristic in our study is likely too low to draw any meaningful conclusions from this observation.
- Was there any association with the remission status, i.e., are patients in PR more at risk for neurotoxic events?
This again is an interesting point raised by the reviewer. We did not find such an association in the LTI study, but a trend in HR-NBL1 towards a higher incidence of severe neurotoxicity in patients who were not in complete response prior to receiving immunotherapy. We added two sentences doe reflect this result (lines 218 and 277).
However, we suspect that our patient numbers are too low to conduct a sound statistical analysis on the correlation of disease status and the frequency of neurotoxic events.
- According to table 3, 16 patients completely recovered, five patients did not and data were missing in six patients. Therefore, the complete recovery rate should better be given as 16/22 instead of 22/27. The same is true for the patient numbers of the respective trials.
We have updated the recovery rate as requested, highlighting the missing data – please see abstract on page 2, lines 184/185 on page 5 and lines 286/287 on page 13 of the revised manuscript. We have also adjusted table 2 on page 5 accordingly and added a footnote to acknowledge the missing data.
- The layout of table 1 should be revised prior to publication in order to improve readability. The target serum concentration of DB should be given.
The layout of table 1 on page 5 of the revised manuscript has now been slightly formatted in order to improve readability. We have also adjusted the formatting of table 3 as we believe the reviewer might have been referring to that table. Please note that we are limited in changing the formatting/layout of tables due to the journal’s template/style. However, we believe that our changes (bullet points etc) have improved the readability of the tables.
We have also added a footnote to table 3 indicating that the expected maximum serum concentration of DB at the end of infusion was 12.56 ± 0.68 µg/ml study as previously reported for the LTI study by Siebert et al. The reference has also been included in the manuscript (highlighted). The mean maximum concentration of DB in patients with severe neurotoxicity was 7.0 µg/ml ± 1.9 µg/ml (range 2.0–14.9 µg/ml), which is lower than the expected level. None of the patients demonstrated an unusually high drug concentration that might explain the neurotoxic side effects. These details have now been added to the results section on (lines 222/223) and the discussion section on page 15 (lines 368-372).
Please note that serum concentrations of DB were only determined in patients included in the LTI study, not the HR-NBL1 study, making it difficult for us to draw conclusions on this issue. However, the increased frequency and severity of neurotoxic events observed in the presence of IL-2 suggests that IL-2 may be the main driver of these events.
- In Table 4, the CSF-examination should be mandatory because CSF is collected for HSV- and HSV-PCR anyway.
We agree with this suggestion and have now updated this recommendation in table 4 (changed from ‘consider investigation of CSF’ to ‘carry out CSF examination’).

Reviewer 2 Report
In the work of Wieczorek et al. a rare but severe neurological side effect of treatment of children with neuroblastoma with an antibody called dinutiuximab beta is described. These antibodies target GD2 which is a disialoganglioside i.e. a sugar/lipid moiety that is preferentially expressed in cancer cells and is administered to patients often in combination with IL-2. Since the pivotal study of Alice L Yu et al. in NEJM 2010 where the anti-GD2 antibody was compared to standard chemotherapy and resulted in considerably better response and 2-year survival rates, more than 10 years have passed. In general, the response to this type of treatment is good and can reduce/substitute the chemotherapeutic burden that may lead to secondary malignancies. In the 2 SIOPEN trials it turned out that these side effects were found exclusively in the patients where IL-2 was added to the antibody, and it is now known that IL-2 does not add to clinical efficacy.
However, in the light of these rare (around 2-5% of cases) but severe side effects one should keep an eye on the neurological risk of this type of treatment. I think the work is well written and an important and new addition to current literature in the field, moreover it provides recommendation how to proceed in case neurological symptoms occur.
Author Response
Thank you very much for the assessment.